# Quantitative Trait Locus Mapping of Seed Vigor in Soybean under −20 °C Storage and Accelerated Aging Conditions via RAD Sequencing

**Rongfan Wang [1], Fengqi Wu [1], Xianrong Xie [2] and Cunyi Yang [1,\*]**

[1] Department of Seed Science and Technology, College of Agriculture, South China Agricultural University, Guangzhou 510642, China; rongfan.wang@outlook.com (R.W.); 360875601w@gmail.com (F.W.)

[2] Department of Genetics, College of Life Sciences, South China Agricultural University, Guangzhou 510642, China; xiexianrong@scau.edu.cn

[\*] Correspondence: ycy@scau.edu.cn; Tel.: +86-20-85280693

**Abstract:** Due to its fast deterioration, soybean (*Glycine max* L.) has an inherently poor seed vigor. Vigor loss occurring during storage is one of the main obstacles to soybean production in the tropics. To analyze the genetic background of seed vigor, soybean seeds of a recombinant inbred line (RIL) population derived from the cross between Zhonghuang24 (ZH24, low vigor cultivar) and Huaxia3hao (HX3, vigorous cultivar) were utilized to identify the quantitative trait loci (QTLs) underlying the seed vigor under −20 °C conservation and accelerated aging conditions. According to the linkage analysis, multiple seed vigor-related QTLs were identified under both −20 °C and accelerated aging storage. Two major QTLs and eight QTL hotspots localized on chromosomes 3, 6, 9, 11, 15, 16, 17, and 19 were detected that were associated with seed vigor across two storage conditions. The indicators of seed vigor did not correlate well between the two aging treatments, and no common QTLs were detected in RIL populations stored in two conditions. These results indicated that deterioration under accelerated aging conditions was not reflective of natural aging at −20 °C. Additionally, we suggest 15 promising candidate genes that could possibly determine the seed vigor in soybeans, which would help explore the mechanisms responsible for maintaining high seed vigor.

**Keywords:** quantitative trait loci (QTLs); soybean (*Glycine max* L.); seed vigor; seed aging; high-density genetic map



## 1. Introduction

Seed vigor is a complex physiological trait, reflecting the comprehensive potentials of seed germination and normal seedling establishment under a wide range of adverse and stressful conditions, such as high temperature and moisture [1]. It is controlled by quantitative genetic factors, the initial condition of the seed, and biotic and abiotic factors during storage [2–4]. The reduction in seed vigor dramatically affects the population density, compromising the grain yield [5–7]. Therefore, maintaining a high seed vigor during post-harvest storage is an essential ingredient for improving crop production, which is of both economic and ecologic importance [8,9]. It is noteworthy that a loss of seed vigor in storage remains problematic for many crops, and this appears particularly severe for oilseed crops such as soybean. Fortunately, there is great variation of seed vigor in plant species including soybean [10–13]. Those cultivars with better seed vigor that are tolerant of adverse conditions during storage would be important germplasm resources for higher soybean yields.

Soybean [*Glycine max* (L.) Merr.], one of the world's most widely planted crops, provides nearly 28% of the vegetable oil and approximately 70% of the dietary protein for humans [14]. China is the largest consumer of soybean, but China's soybean production falls far short of its increasing demand. In 2019, imports of soybean into China were about

88.51 million tons, accounting for over 60% of global imports [15]. Hence, to alleviate the pressure of production shortfall in soybean, the Chinese government proposed the expansion of planting areas in the subtropics and tropics for soybean. However, a major constraint in subtropical and tropical soybean production arises from the deterioration of the seeds in storage, since soybean seeds have an inherently short life span compared to other crop species [4]. Accordingly, the major thrust of soybean breeding in tropical regions is to develop cultivars with superior seed vigor levels. However, despite the fact that seed vigor plays a key role in soybean production, few relevant studies on the genetic analysis of soybean have been reported so far.

Seed vigor is a complex trait controlled by multiple genes. The use of molecular markers has provided insights into the genomic location and gene action of quantitative trait loci (QTLs) for seed vigor during soybean storage. Up to the present, 18 QTLs associated with seed vigor have been identified on 15 chromosomes using different soybean populations [16–19]. According to the germination percentage of $F_{2:3}$ progenies from 'Birsa soya-1' (black-seeded variety with long lifespan) and 'JS71-05' (yellow-seeded variety with short longevity) subjected to accelerated aging (AA) tests in the laboratory, Singh et al. [20] detected four SSR markers (Satt600, Satt538, Satt434, and Satt285 located on chromosome 2, 8, 12, and 16, respectively) associated with seed vigor. In addition, based on the seed germination performance of 33 soybean genotypes under ambient and AA storages, three SSR markers (Satt371, Satt453, and Satt618) were identified for linkage with both seed vigor and seed coat color [19]. Dargahi et al. [17] mapped three independent QTLs on chromosomes 4, 13, and 19 via the relative germination rate in the $F_3$ and $F_4$ populations from a cross between 'MJ0004-6' with poor longevity and 'R18500' with good longevity, but a common QTL for seed vigor was missing from the seeds produced in different years. Using high-density linkage mapping by two recombinant inbred line (RIL) populations ('Zhengyanghuangdou' × 'Meng 8206' and 'Linhefenqingdou' × 'Meng 8206'), Zhang et al. [18] identified one novel 'QTL hotspot region' for seed vigor under natural aging (NA) and AA conditions, which lay between 39,676,735 and 41,073,260 on chromosome 17. To date, there is no consistent QTL of seed vigor in soybean across diverse genetic populations. It is therefore necessary to explore a wide range of deterioration-tolerant germplasms to develop tropically adapted cultivars that meet the needs of farmers.

To investigate vigor levels in seeds, accelerated aging (AA) or controlled deterioration (CD) treatments using high temperatures and a high relative humidity are mostly employed by exacerbating post-harvest deterioration [11,21,22]. Under such storage conditions, seeds typically lose viability within a few days or weeks, while seeds can age over many years under cold dry storage. Hence, some researchers have assumed that the major primary process that initiates seed aging could be different under AA and NA conditions, and the controversy on whether mechanisms of seed aging are similar under different deterioration treatments continues. During deterioration, a range of irreversible metabolic and cellular alterations including the oxidation of lipids, proteins and nucleic acids, enzyme inactivation, membrane perturbations, and impairment of DNA, RNA and protein biosynthesis generally occur in aged seeds [2,4,23]. In this regard, the levels of seed vigor are considered to be associated with the balance between oxidative damages and self-protective as well as repair mechanisms such as antioxidant systems [24]. Profiling genetic analysis in soybean seeds under different aging conditions may allow for a better understanding of the mechanism of seed vigor and a better understanding of how to enhance the monitoring of seed deterioration in long-term germplasm conservation [25].

In this study, we used a RIL population developed from a cross between two diploid cultivars (cv.) 'Zhonghuang24' and 'Huaxia3hao'. Via restriction site-associated DNA sequencing (RAD-seq), a high-density genetic linkage map was obtained to map the QTLs for the seed vigor of soybean under −20 °C and accelerated aging conditions. This study aimed to detect QTLs across different storage environments and to select reliable candidate genes in these genetic intervals for the seed vigor of soybean.

## 2. Materials and Methods

### 2.1. Plant Materials and Storage Conditions

A RIL mapping population consisting of 168 progenies was constructed by the cross of soybean cv. ZH24 (low vigor cultivar) and cv. HX3 (vigorous cultivar). 'ZH24' is a cultivar adaptive to Huang-Huai-Hai Rivers Valley China, while 'HX3' is a high-yielding variety which was obtained from by South China Agriculture University [26]. The RIL populations ($F_{7–9}$) and the parents were planted during summer (from June to November) at Zengcheng, Guangzhou (N 23°24′, E 113°64′), South China in 2013 and 2018. Each 1 m long row with 10–15 plants and 0.5 m space between rows were arranged following a randomized block design. The seeds harvested in 2013 were preserved in sealed bags at −20 °C until use, while seeds harvested in 2018 were kept under ambient conditions before use. The site details, climate, and storage conditions are well described in Table S1.

### 2.2. Evaluation of Seed Vigor

The germination ability of seeds harvested in 2013 from ZH24, HX3, and all RILs was determined after 5 years of storage at −20 °C (naturally aged), while the seeds from 2018 were artificially aged before the germination test. The germination experiment and accelerated aging treatment were performed according to the International Rules for Seed Testing [1]. For accelerated aging, the seed moisture content was calibrated to 10–14%, then half of them were placed in the aging chamber (41 °C, relative humidity 99%) for 72 h under dark conditions. Afterward, three replicates with 20 seeds each for all the materials were planted in sterilized vermiculite in the climate chamber for 7 days (14 h/10 h light length, 25 °C). The numbers of germinated seeds were counted daily, and the numbers of normal seedlings, fresh weight per seedling, shoot length as well as root length were recorded on the 7th day. The parameters related to seed vigor were calculated based on the following equations:

Germination potential (GP) = $m/N \times 10$, where m is the number of germinated seeds on the third day, and N is the total number of seeds. Germination rate (GR) was calculated as GR = $n/N \times 100$, where n is the total number of germinated seeds on the 7th day. Normal seedling rate (NSR) was calculated as NSR = $r/N \times 100$, where r is the total number of normal growth seedlings. The simple vigor index (SVI) was calculated as SVI = $GR \times SFW$, where SFW is the seedling fresh weight on the 7th day after seed germination. Germination index (GI) was calculated as GI = $\Sigma(Gt/Dt)$, where Dt is the germination time, and Gt is the number of germinated seeds during the germination time. Vigor index (VI) was calculated as VI = $GI \times SFW$.

### 2.3. SNP Genotyping

Based on 30× whole genome sequencing (WGS) of parental lines and 0.2× restriction site-associated DNA sequencing (RAD-seq) of $F_6$ RILs, the Burrows–Wheeler Aligner was used to align the clean reads of each sample against the soybean reference genome (Williams 82 Assembly 2 Genomic Sequence, Wm82.a2.v1) (https://www.soybase.org/, (accessed on 12 January 2014)). Variant calling was performed using GATK for parents and freebayes for RILs, respectively, then single nucleotide polymorphisms (SNPs) and Insertion/Deletion (InDels) of RILs were filtered according to polymorphic parental markers with proper standards. Besides, heterozygous markers, variants outside the sequencing depth range of 4–1000, and variant sites with the missing proportion higher than 0.25 and with a poor sequencing quality or those exhibiting a significantly distorted segregation were discarded. The RAD-seq data were been also utilized to analyze the genetic basis of yield-related and two quality traits [27], long juvenile trait [28], and leaf type traits [26] for soybean. WGS was conducted by Annoroad Gene Technology, Beijing, China, and RAD-seq was carried out at the Beijing Genome Institute (BGI) Tech, Shenzhen, China.

To assess the sequence variants and their impact, the effects of sequence ontology (SO) were categorized into high, moderate, low, and modifier with detailed descriptions

predicted by SnpEff (http://snpeff.sourceforge.net/SnpEff_manual.html, (accessed on 12 January 2015)).

### 2.4. High-Density Linkage Map Construction and QTL Analysis

To overcome the false positive of the SNPs genotype of the population, the binning of redundant markers was developed by QTL IciMapping version 4.2 [29] (http://www.isbreeding.net, (accessed on 24 July 2019)). The qualified bin markers were used to construct the genetic linkage map via Joinmap 4 [30] and MapChart 2.32 [31]. The inclusive composite interval mapping (ICIM) method was employed to detect QTLs [29]. The threshold for the logarithm of odds (LOD) scores declaring a significant additive QTL was set as 2.5 using LOD Threshold manual input: ICIM-ADD. QTLs were named according to Mccouch et al. [32] and QTL mapping results were comprehensively compared to previous reports.

### 2.5. GO Enrichment Analysis

To find the most promising candidate genes, the variants of genes with low and modifier effects were filtered. Then, the regional genes of detected QTLs were annotated and analyzed via the Soybase database (http://www.soybase.org/, (accessed on 2 December 2020)). According to the properties of genes and their products in organisms, all candidate genes were mapped to three Gene Ontology (GO) terms: molecular function (MF), cellular component (CC), and biological process (BP) in the database (http://www.geneontology.org/, (accessed on 18 August 2021)).

### 2.6. Statistical Analysis

The mean values of all the phenotypic data obtained for two parental cultivars (ZH24, HX3) and their RIL populations were utilized for a Student's t-test and descriptive statistics using SPSS Statistics 22.0 (SPSS, Inc., Chicago, IL, USA). *, **, and *** indicate significant differences at the 0.5, 0.01, and 0.001 probability levels, respectively. The frequency histograms of the seed vigor indices for RILs were generated using GraphPad Prism 8 (Version 8.03, GraphPad Software, San Diego, CA, USA).

Spearman's rank correlations for all traits between two storage conditions were analyzed based on the phenotypic values of all the RILs. *, **, and *** indicate significant correlations at $p$-values below 0.5, 0.01, and 0.001, respectively.

## 3. Results

### 3.1. Phenotypic Performance of the Parents and RIL Populations for Seed Vigor

Nine parameters, including germination potential (GP), germination rate (GR), normal seedling rate (NSR), germination index (GI), shoot length (SL), root length (RL), seedling fresh weight (SFW), vigor index (VI), and simple vigor index (SVI) of each RIL and their parents under −20 °C storage and accelerated aging condition were studied to evaluate seed vigor (Table 1). Compared with cv. Zhonghuang24 (ZH24), cv. Huaxia3hao (HX3) exhibited significantly higher values in most of the traits after aging treatments, indicating that HX3 had better vigor than ZH24. After 5 years of storage at −20 °C, the germination rate and root length were rarely affected by seed aging, while the effects on shoot length and seedling fresh weight were remarkably different between the parents. By contrast, accelerated aging induced a poor germination rate and root development in the less vigorous cultivar (ZH24), but shoot development and SFW were nearly unaffected. According to the observations, naturally aged seeds from ZH24 produced severe damage in respect of seedling establishment, whereas the germination capacity under the artificially aged conditions was significantly reduced, suggesting seed responses to aging varied across different storage conditions as well.

**Table 1.** Phenotypic performance of ZH24, HX3 and their 168 $F_{7-9}$ RILs for seed vigor under natural and accelerated aging conditions.

| Traits [1] | Treatments [2] | Parental Lines | | RILs [3] | | | | |
|---|---|---|---|---|---|---|---|---|
| | | ZH24 | HX3 | Mean ± SD | Range | CV (%) | Kurtosis | Skewness |
| SL (cm) | NA | 7.3 ± 0.6 * | 9.4 ± 0.5 | 9.4 ± 1.0 | 5.9–11.6 | 10.83 | 0.34 | −0.44 |
| | AA | 8.4 ± 0.4 | 8.5 ± 0.8 | 8.5 ± 1.0 | 5.7–12.3 | 11.81 | 1.16 | 0.66 |
| RL (cm) | NA | 17.1 ± 1.0 | 16.8 ± 1.4 | 17.3 ± 1.8 | 12.9–24.9 | 10.24 | 1.26 | 0.47 |
| | AA | 13.7 ± 2.5 ** | 19.5 ± 1.1 | 19.1 ± 2.8 | 9.3–26.3 | 14.63 | 0.50 | −0.32 |
| GP (%) | NA | 23.0 ± 16.0 * | 65.0 ± 18.0 | 14.3 ± 13.3 | 0.0–68.3 | 92.8 | 3.09 | 1.60 |
| | AA | 10.8 ± 3.0 ** | 60.8 ± 7.4 | 59.8 ± 21.9 | 3.3–100.0 | 36.55 | −0.74 | −0.29 |
| GR (%) | NA | 95.0 ± 5.0 | 100.0 ± 0.0 | 99.1 ± 2.9 | 68.3–100.0 | 2.95 | 75.12 | −7.66 |
| | AA | 55.0 ± 13.4 ** | 95.8 ± 8.0 | 89.6 ± 9.6 | 46.7–100.0 | 10.7 | 2.40 | −1.38 |
| NSR (%) | NA | 85.0 ± 8.7 * | 100.0 ± 0.0 | 95.7 ± 6.4 | 53.3–100.0 | 6.73 | 16.11 | −3.53 |
| | AA | 39.2 ± 8.6 ** | 90.8 ± 9.7 | 81.8 ± 13.8 | 16.7–100.0 | 16.89 | 3.00 | −1.31 |
| GI | NA | 15.0 ± 1.9 * | 19.5 ± 1.2 | 15.6 ± 1.6 | 7.5–20.0 | 10.08 | 4.91 | −0.81 |
| | AA | 8.2 ± 1.6 ** | 18.0 ± 2.3 | 17.2 ± 2.8 | 7.7–23.0 | 16.33 | 0.07 | −0.55 |
| SFW (g) | NA | 0.897 ± 0.086 *** | 1.426 ± 0.036 | 1.272 ± 0.209 | 0.675–1.834 | 16.39 | 0.01 | 0.04 |
| | AA | 1.163 ± 0.187 | 1.248 ± 0.136 | 1.321 ± 0.236 | 0.812–2.078 | 17.82 | 0.18 | 0.51 |
| SVI | NA | 0.855 ± 0.126 ** | 1.426 ± 0.036 | 1.262 ± 0.215 | 0.646–1.834 | 17.01 | 0.07 | −0.05 |
| | AA | 0.630 ± 0.128 ** | 1.199 ± 0.185 | 1.192 ± 0.286 | 0.611–2.008 | 23.98 | −0.38 | 0.34 |
| VI | NA | 13.398 ± 1.248 *** | 27.860 ± 2.431 | 19.831 ± 3.746 | 8.471–29.273 | 18.88 | 0.52 | −0.27 |
| | AA | 9.408 ± 1.694 ** | 22.455 ± 3.094 | 22.926 ± 6.414 | 10.011–41.985 | 27.97 | −0.45 | 0.28 |

[1] SL, Shoot length; RL, Root length; GP, Germination potential; GR, Germination rate; NSR, Normal seedling rate; GI, Germination index; SFW, Seedling fresh weight; SVI, Simple vigor index; VI, Vigor index; [2] NA, Natural aging; AA, Accelerated aging; [3] RILs, Recombinant inbred lines; SD, Standard deviation; CV, Coefficient of variation; *, **, and *** indicate significant difference at the 0.5, 0.01 and 0.001 probability level, respectively.

The coefficient variance (CV) of the phenotypic values ranged from 10.8 to 11.8% (SL), 10.2 to 14.6% (RL), 16.4 to 17.8% (SFW), 21.2 to 92.8% (GP), 3.0 to 10.7% (GR), 6.7 to 16.9% (NSR), 17.0 to 24.0% (SVI), 10.1 to 16.3% (GI), and 18.9 to 28.0% (VI), respectively, under different conditions. Meanwhile, the skewness and kurtosis of these measured traits suggest that most traits fit the normal distribution except GR and NSR under natural aging conditions. The considerable variation and continuous distributions of the phenotypic values for the RIL populations (Figure 1) indicate that soybean seed vigor is a typical quantitative genetic characteristic.

Different extents of positive correlation were detected among the nine evaluated parameters under the same storage conditions (Table S2 and Figure 2). Specifically, the correlation coefficients of the germination-related parameters (GP and GI) as well as the seedling-related indicators (SFW, SVI, and VI) were between 0.84 and 0.95, 0.84 and 0.99, respectively, which suggests that GP and GI were significantly correlated with each other under the same conditions, as well as SFW, SVI, and VI. All germination-related parameters (GP, GP, GI, and NSR) were found to be highly correlated with one another under NA and AA conditions. However, the values of the measured indicators from naturally aged seeds were poorly correlated with those of the artificially aged seeds (r = −0.11 to r = 0.38). The correlation analyses revealed that a combination of genetic and environmental factors determines the seed vigor for soybean.

*3.2. Identification of the QTL Determining the Seed Vigor of Soybean*

Based on the alignments of ZH24 and HX3 with the reference genome, a total of 1,729,014 polymorphic SNPs and InDels between two parents were detected, and 52,306 of them were also found for the RILs. Based on 0.2 × RAD-seq of the RIL population, a total of 52306 SNP sites/markers were detected. Since some markers were redundant, meaning that they were completely correlated or identical in the population, they were integrated into a recombination bin unit. After further filtration and analysis, 5425 recombinant

bins were obtained from high-quality polymorphic SNP sites in the RILs for our study (Figure S1). According to the genotypic data at the bin marker loci for all individuals in the population, a high-density linkage map with a total length of 7932.8 cM was constructed. The numbers of the bin markers on each chromosome ranged from 145 to 402, and the average genetic distance between the adjacent bins was 1.46 cM (Table S3).

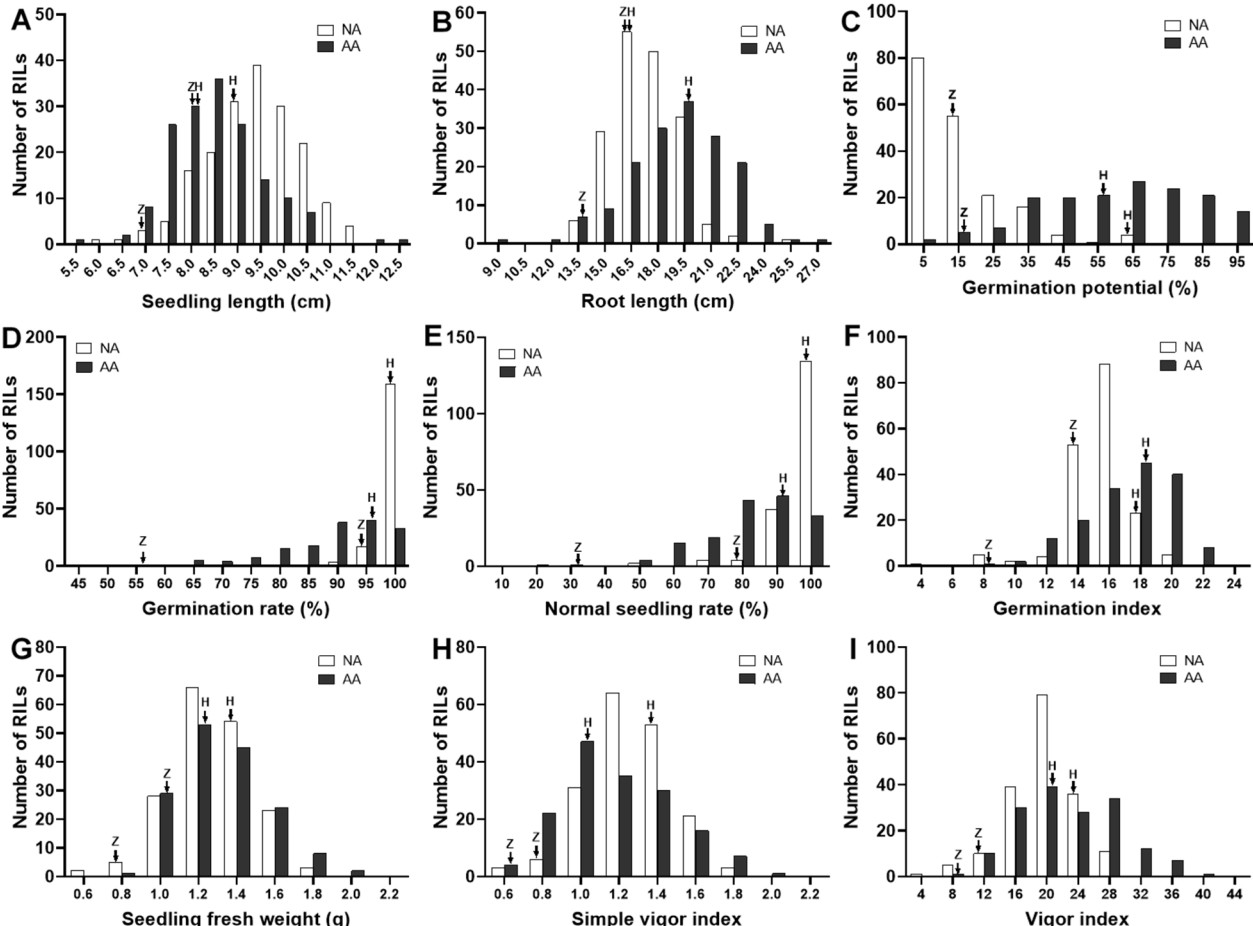

**Figure 1.** Frequency distribution of the phenotypic values for seedling length (**A**), root length (**B**), germination potential (**C**), germination rate (**D**), normal seedling rate (**E**), germination index (**F**), seedling fresh weight (**G**), seed vigor index (**H**), and vigor index (**I**) for the 168 RILs stored under −20 °C and accelerated aging conditions, respectively. The arrows indicate the trait-related values for the two parents used to construct the RIL populations (Z, cv. Zhonghuang24; H, cv. Huaxia3hao). NA, natural aging; AA, accelerated aging.

On the basis of the linkage map, an association analyses with phenotypic data detected 48 QTLs for seed vigor under two aging conditions, among which 30 QTLs for −20 °C storage and 18 for accelerated aging were identified (Table 2, Figure 3). Among them, 18 loci with positive additive effects were derived from the alleles of the male parent (HX3), whereas those showing negative additive effects were derived from the female parent (ZH24). These loci explained 2.33–21.74% of the phenotypic variance with LOD values from 2.51 to 19.92. Although several QTL hotspots presented within the same storage condition, the overlap of QTLs was absent between different conditions, indicating a strong genotype of seed vigor by environmental interactions. By a comparison with published QTLs in previous research, ten seed vigor-related loci from six chromosomes (3, 6, 8, 9, 12, and 13) detected in our study were also responsible for other traits in different soybean cultivars (Table 2).

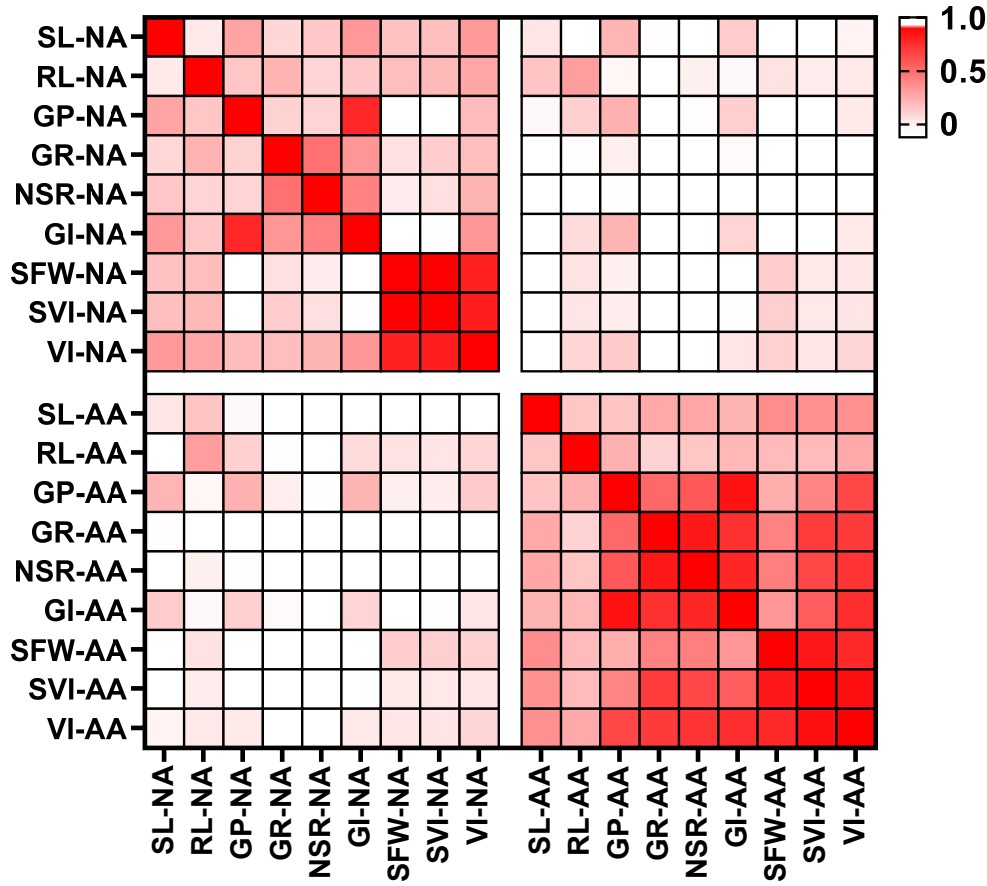

**Figure 2.** Correlation matrix of Spearman's rank correlation coefficients between all phenotypic traits under different aging conditions. The correlation coefficient was calculated based on the phenotypic values of all the RILs, and the correlation values are shown in Table S2. Red squares show positive correlations whereas the white squares show poor correlations. SL, shoot length; RL, root length; SFW, seedling fresh weight; GP, germination potential; GR, germination rate; NSR, normal seedling rate; SVI, simple vigor index; GI, germination index; VI, vigor index; NA, natural aging; AA, accelerated aging.

**Table 2.** The characteristics of QTLs associated with seed vigor in the RILs under natural and accelerated aging conditions.

| QTLs | Traits and Treatments [1] | Chr. [2] | Marker Interval (cM) | Physical Interval (bp) | LOD [3] | PVE [4] (%) | ADD [5] | Published Loci [6] |
|---|---|---|---|---|---|---|---|---|
| *qRL-1* | RL-NA | 1 | 248.5–249.5 | 2,266,063–2,275,220 | 3.48 | 13.01 | 0.77 | |
| *qSL-3* | SL-AA | 3 | 90.5–96.5 | 41,574,342–41,662,968 | 3.35 | 7.45 | 0.29 | |
| *qRL-3* | RL-NA | 3 | 159.5–161.5 | 35,272,823–35,406,160 | 3.01 | 4.26 | −0.44 | *qFe-3* [33]; maximum root length, shoot weight [34] |
| *qGP-3* | GP-AA | 3 | 269.5–272.5 | 9,236,943–18,541,535 | 3.04 | 2.56 | −0.14 | *qFe-3* [33]; maximum root length, shoot weight [34] |
| *qGI-3* | GI-AA | 3 | 267.5–272.5 | 9,236,943–18,541,535 | 2.56 | 12.04 | −1.32 | *qFe-3* [33]; maximum root length, shoot weight [34] |
| *qSVI-4* | SVI-NA | 4 | 413.5–414 | 11,602,229–11,625,478 | 2.60 | 14.10 | −0.13 | |
| *qRL-5* | RL-NA | 5 | 38.5–40.5 | 422,059–725,212 | 4.09 | 5.85 | 0.51 | |
| *qSL-6.1* | SL-NA | 6 | 86.5–89.5 | 763,415–834,819 | 2.53 | 6.03 | 0.26 | |
| *qSL-6.2* | SL-NA | 6 | 229.5–234.5 | 11,656,787–12,013,785 | 4.50 | 12.11 | −0.37 | |
| *qGP-6* | GP-AA | 6 | 278.5–279.5 | 14,814,912–15,030,638 | 2.80 | 3.74 | −0.17 | |
| *qGR-6* | GR-AA | 6 | 46.5–51.5 | 1,847,293–2,107,875 | 2.53 | 8.77 | 0.03 | |
| *qNSR-6* | NSR-AA | 6 | 46.5–52.5 | 1,847,293–2,107,875 | 2.80 | 9.60 | 0.04 | |
| *qGI-6* | GI-AA | 6 | 46.5–52.5 | 1,847,293–2,107,875 | 3.05 | 5.62 | 0.92 | |
| *qSFW-6.1* | SFW-NA | 6 | 229.5–234.5 | 11,656,787–12,013,785 | 2.94 | 4.29 | −0.06 | |
| *qSFW-6.2* | SFW-NA | 6 | 455.5–456.5 | 21,944,001–33,594,563 | 2.86 | 20.15 | −0.13 | protein content [35]; hypocotyl weight [34] |
| *qSVI-6.1* | SVI-NA | 6 | 226.5–229.5 | 12,013,785–12,031,741 | 3.27 | 3.29 | −0.07 | |
| *qSVI-6.2* | SVI-NA | 6 | 455.5–456.5 | 21,944,001–33,594,563 | 2.55 | 13.68 | −0.13 | protein content [35]; hypocotyl weight [34] |
| *qVI-6* | VI-NA | 6 | 226.5–235.5 | 11,656,787–12,013,785 | 5.07 | 13.35 | −1.42 | |
| *qRL-7* | RL-NA | 7 | 39.5–41.5 | 38,975,809–39,197,054 | 2.82 | 10.49 | −0.70 | |
| *qSL-8* | SL-NA | 8 | 321.5–328.5 | 5,465,698–5,763,658 | 2.91 | 7.12 | −0.28 | *qSTGI-8, qSTGP-8-1, qSTGR-8* [36] |
| *qRL-8* | RL-NA | 8 | 49.5–52.5 | 44,220,876–44,360,808 | 5.38 | 7.86 | −0.60 | |
| *qSL-9* | SL-NA | 9 | 73.5–75.5 | 3,277,067–3,447,421 | 3.51 | 8.52 | −0.30 | |

**Table 2.** *Cont.*

| QTLs | Traits and Treatments [1] | Chr. [2] | Marker Interval (cM) | Physical Interval (bp) | LOD [3] | PVE [4] (%) | ADD [5] | Published Loci [6] |
|---|---|---|---|---|---|---|---|---|
| *qRL-9* | RL-NA | 9 | 237.5–238.5 | 34,536,745–34,592,783 | 6.39 | 9.55 | 0.66 | seed oil concentration [37] |
| *qGP-9* | GP-NA | 9 | 95.5–97.5 | 4,829,938–4,923,325 | 3.66 | 11.65 | −0.04 | |
| *qGI-9* | GI-NA | 9 | 95.5–97.5 | 4,829,938–4,923,325 | 8.36 | 7.55 | −0.68 | |
| *qVI-9* | VI-AA | 9 | 394.5–396.5 | 45,606,441–46,125,101 | 2.71 | 6.83 | −4.40 | |
| *qSFW-11* | SFW-NA | 11 | 178.5–185.5 | 7,422,403–7,439,005 | 4.10 | 5.70 | −0.07 | |
| *qSVI-11* | SVI-NA | 11 | 178.5–185.5 | 7,422,403–7,439,005 | 5.06 | 4.98 | −0.08 | |
| *qVI-11* | VI-NA | 11 | 178.5–184.5 | 7,422,403–7,439,005 | 4.62 | 11.33 | −1.30 | |
| *qGI-12* | GI-NA | 12 | 186.5–215.5 | 34,335,331–36,411,634 | 2.57 | 2.85 | −0.42 | hypocotyl weight [34]; *GR1* [38] |
| *qSL-13* | SL-AA | 13 | 281.5–283.5 | 23,865,825–23,893,516 | 2.58 | 5.64 | −0.25 | |
| *qGP-13* | GP-AA | 13 | 114.5–115.5 | 37,087,514–37,202,638 | 2.55 | 3.65 | 0.17 | *qMg-13* [33] |
| *qNSR-13* | NSR-NA | 13 | 103.5–105.5 | 38,038,412–38,158,435 | 2.76 | 9.76 | 0.02 | *qMg-13* [33] |
| ***qGI-15.1*** | **GI-NA** | **15** | **54.5–56.5** | **45,679,079–46,440,795** | **19.92** | **21.71** | **1.15** | |
| *qGI-15.2* | GI-NA | 15 | 98.5–101.5 | 26,272,123–31,027,649 | 11.65 | 10.99 | 0.82 | |
| *qGP-16* | GP-AA | 16 | 261.5–262.5 | 8,064,570–8,093,843 | 3.11 | 3.88 | −0.18 | |
| *qNSR-16* | NSR-AA | 16 | 358.5–364.5 | 781,601–793,493 | 2.54 | 7.85 | −0.04 | |
| *qSVI-16* | SVI-AA | 16 | 227.5–228.5 | 21,027,685–25,351,732 | 3.02 | 8.59 | −0.21 | |
| *qVI-16* | VI-AA | 16 | 227.5–228.5 | 21,027,685–25,351,732 | 3.17 | 8.33 | −4.84 | |
| *qGP-17* | GP-AA | 17 | 71.5–73.5 | 4,938,743–5,130,517 | 2.51 | 3.05 | 0.16 | |
| *qGI-17* | GI-NA | 17 | 184.5–189.5 | 26,308,785–27,276,853 | 2.87 | 2.33 | 0.38 | |
| *qSFW-17* | SFW-NA | 17 | 130.5–134.5 | 11,412,903–11,464,476 | 5.10 | 7.19 | 0.08 | |
| *qSVI-17* | SVI-NA | 17 | 130.5–134.5 | 11,412,903–11,464,476 | 5.33 | 5.25 | 0.08 | |
| *qVI-17* | VI-NA | 17 | 128.5–131.5 | 11,412,903–11,442,354 | 4.73 | 11.69 | 1.31 | |
| ***qSL-19*** | **SL-AA** | **19** | **129.5–132.5** | **41,469,007–41,759,276** | **5.68** | **13.15** | **−0.38** | |
| *qGI-19* | GI-NA | 19 | 153.5–156.5 | 40,448,911–40,566,340 | 4.81 | 4.00 | −0.50 | |
| *qSL-20* | SL-AA | 20 | 395.5–401.5 | 540,229–611,070 | 3.94 | 9.01 | 0.32 | |
| *qRL-20* | RL-AA | 20 | 38.5–40.5 | 45,491,490–45,500,896 | 2.64 | 9.24 | −0.84 | |

[1] SL, Shoot length; RL, Root length; SFW, Seedling fresh weight; GP, Germination potential; GR, Germination rate; NSR, Normal seedling rate; SVI, Simple vigor index; GI, Germination index; VI, Vigor index; NA, Natural aging; AA, Accelerated aging; [2] Chr., Chromosome; [3] LOD, Logarithm of odds; ADD, Additive effect; a positive value indicates the superiority of HX3. [4] PVE (%), Percentage of phenotypic variation explained (%); [5] STGI, STGP, STGR indicate GI, GP, GR values under salt stress; [6] QTL names based on soybase.org and previous reports. QTLs marked in bold are major QTLs identified under two aging conditions.

Under the natural aging conditions, five sets of QTLs (*qSL-6.2*, *qSFW-6.1* and *qVI-6*; *qSFW-6.2* and *qSVI-6.1*; *qGP-9* and *qGI-9*; *qSFW-11*, *qSVI-11* and *qVI-11*; *qSFW-17*, *qSVI-17* and *qVI-17*) overlapped, which further illustrates the high correlation between these traits (Table 3). Noticeably, one strong major QTL (*qGI-15.1*), associated with germination potential explained 21.71% of the phenotypic variation (Table 2). The physical position of *qGI-15.1* is located in an interval of 45,679,079–46,440,795 bp on chromosome 15. Moreover, the seeds produced in 2018 under accelerated aging conditions showed three significant vigor-related QTL hotspots (*qGP-3* and *qGI-3*; *qGR-6*, *qNSR-6* and *qGI-6*; *qSVI-16* and *qVI-16*) on chromosomes 3, 6, and 16 that collectively explained 55.51% of the phenotypic variation (PVE). The major QTL *qSL-19* (PVE at 13.15%) detected in artificially aged seeds was located in the region between 41,469,007 and 41,759,276 of chromosome 19 (Tables 2 and 3).

### 3.3. Gene Ontology Enrichment Analysis within Major QTLs and QTL Hotspots

The effects of genetic variants within the intervals of identified QTLs between parents ZH24 and HX3 were predicted by SnpEff. According to the manual for SnpEff version 4.0, a total of 32,616 polymorphism SNPs and Indels were categorized into four types of impact (high, moderate, low, and modifier) (Tables S4 and S5). As shown in Figure 4, we found that only 3.4% of all variant genes (16 of 465 genes) had high potential to produce a phenotypic difference. Genetic variants in two major QTLs (*qGI-15.1* and *qSL-19*) in SO terms included

the missense variant, the synonymous variant splice region variant, and the intergenic region, etc., which were classified into moderate, low, or modifier types, indicating less significant impacts on gene functions (Table S4).

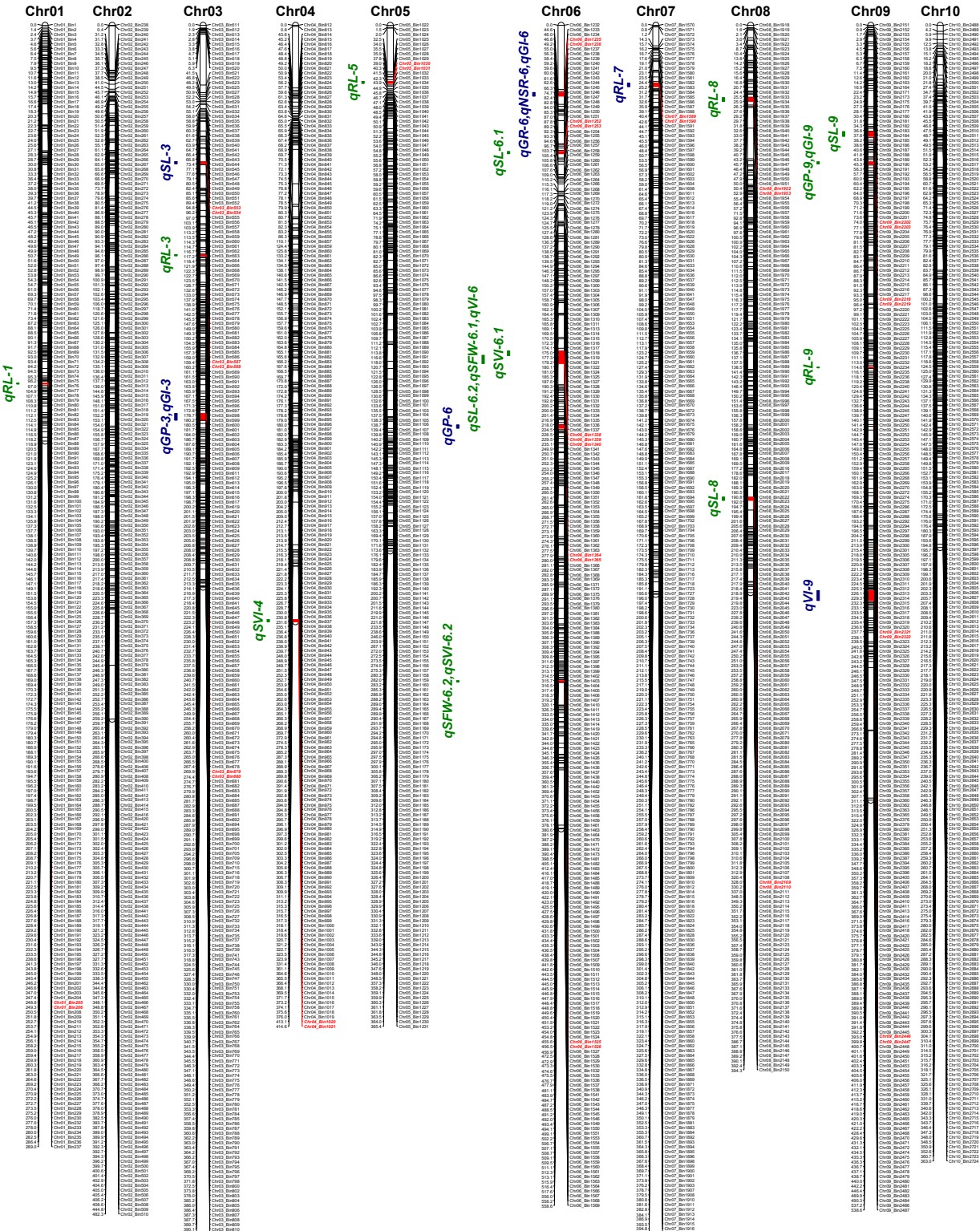

**Figure 3.** *Cont.*

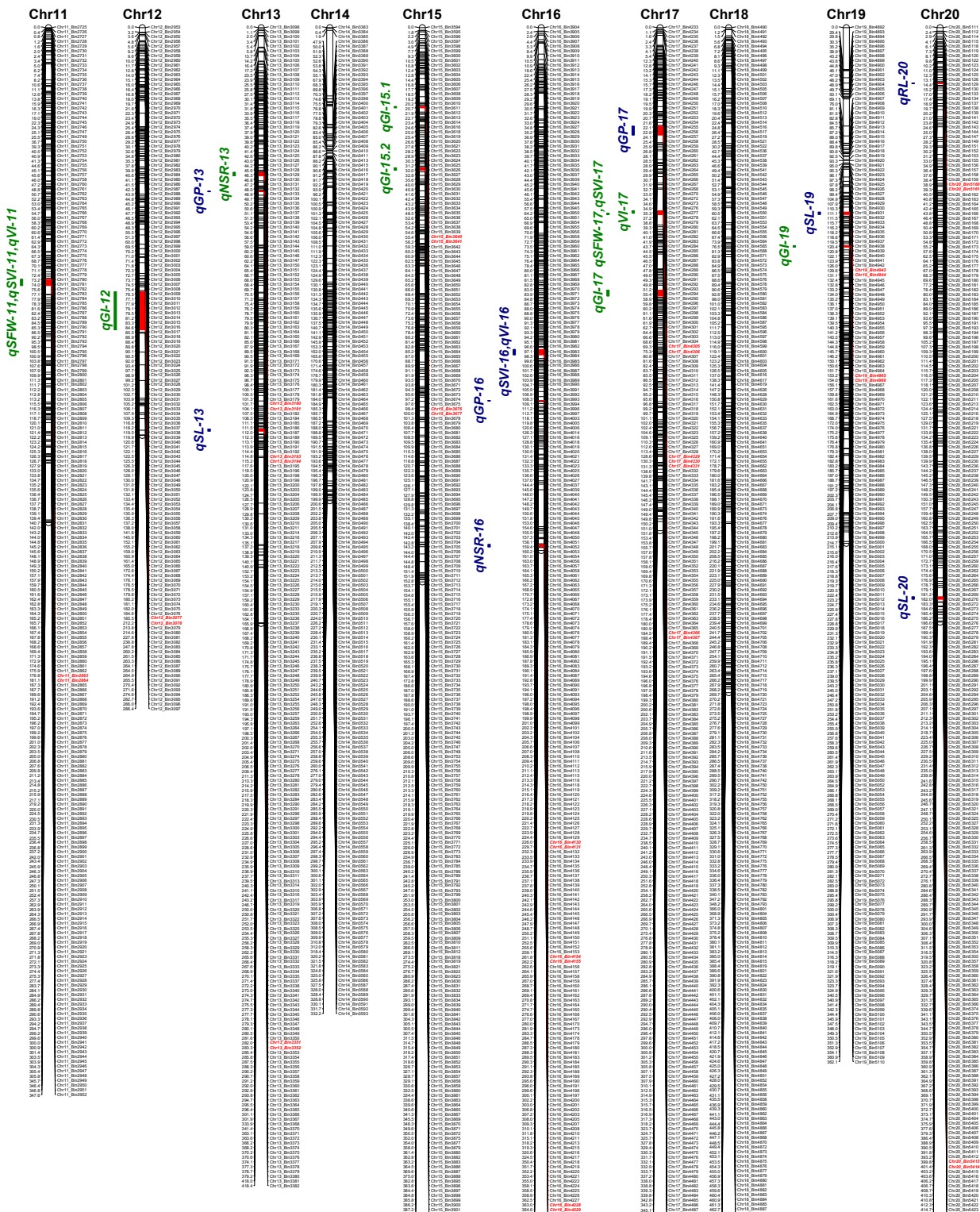

**Figure 3.** The position of the QTLs for seed vigor of soybean on 20 chromosomes. The bin markers and their locations are shown on the right and left sides, respectively. The loci for seed vigor-associated traits are marked and highlighted in red. QTLs labeled in green were detected under natural aging conditions, and the blue color denotes QTLs found under accelerated aging conditions. Map distances are shown in cM. Chr, chromosome.

**Table 3.** The major QTLs and QTL hotspots for seed vigor in the RIL populations across natural and accelerated aging conditions.

| | Loci | QTLs [1] | Chr. [2] | Marker Interval (cM) | Physical Interval (bp) |
|---|---|---|---|---|---|
| Major QTLs | Loci1 | *qGI-15.1* (NA); | 15 | 54.5–56.5 | 45,679,079–46,440,795 |
| | Loci2 | *qSL-19* (AA); | 19 | 129.5–132.5 | 41,469,007–41,759,276 |
| Hotspots | Loci3 | *qGP-3* (AA); *qGI-3* (AA); | 3 | 269.5–272.5 | 9,236,943–18,541,535 |
| | Loci4 | *qGR-6* (AA); *qNSR-6* (AA); *qGI-6* (AA); | 6 | 46.5–52.5 | 1,847,293–2,107,875 |
| | Loci5 | *qSL-6.2* (NA); *qSFW-6.1* (NA); *qVI-6* (NA); | 6 | 229.5–234.5 | 11,656,787–12,013,785 |
| | Loci6 | *qSFW-6.2* (NA); *qSVI-6.1* (NA); | 6 | 455.5–456.5 | 21,944,001–33,594,563 |
| | Loci7 | *qGP-9* (NA); *qGI-9* (NA); | 9 | 95.5–97.5 | 4,829,938–4,923,325 |
| | Loci8 | *qSFW-11* (NA); *qSVI-11* (NA); *qVI-11* (NA); | 11 | 178.5–185.5 | 7,422,403–7,439,005 |
| | Loci9 | *qSVI-16* (AA); *qVI-16* (AA); | 16 | 227.5–228.5 | 21,027,685–25,351,732 |
| | Loci10 | *qSFW-17* (NA); *qSVI-17* (NA); *qVI-17* (NA) | 17 | 130.5–134.5 | 11,412,903–11,464,476 |

[1] GI, Germination index; GP, Germination potential; GR, Germination rate; NSR, Normal seedling rate; SFW, Seedling fresh weight; SL, Shoot length; SVI, Simple vigor index; VI, Vigor index; NA, Natural aging; AA, Accelerated aging; [2] Chr., Chromosome.

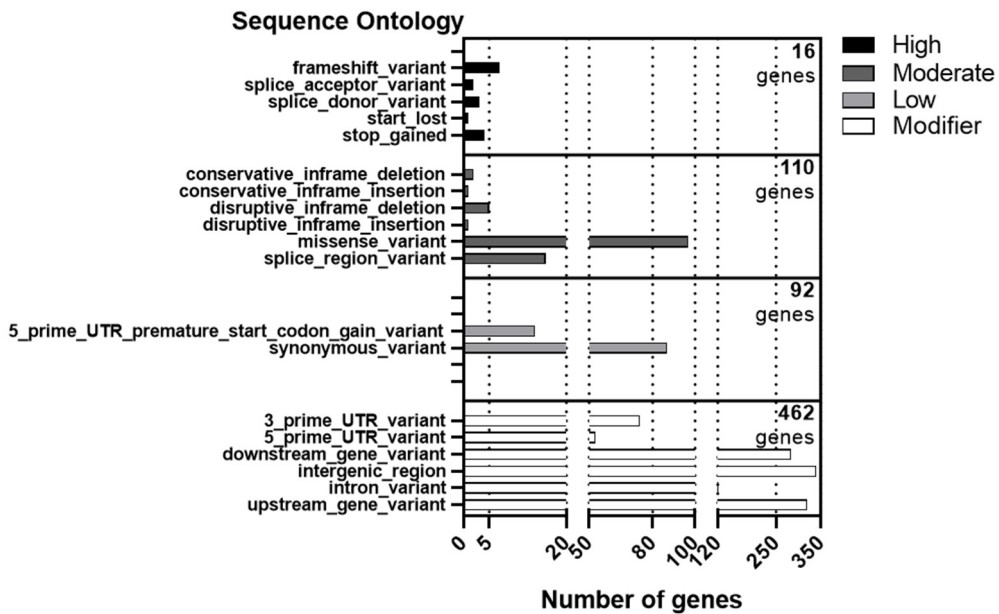

**Figure 4.** The classification of genetic variants within the intervals of two major QTLs and QTL hotspots. The sequence variants between parents ZH24 and HX3 are classified into four types (high, moderate, low, and modifier) according to the effect prediction by SnpEff. The number of variant genes is shown in each category.

*3.4. Candidate Gene Selection*

To find the most promising candidate genes for seed vigor based on GO analysis, we focused on genetic variants belonging to high and moderate types for further study. In



total, 5 genes in two major QTLs and 126 genes in QTL spots were involved in various cellular components, molecular functions, and biological processes, including 21 genes without annotation in public databases (Tables S6 and S7). Among the 67 genes functioning in cellular components, 50 are related to the membrane, nucleus, and cytoplasm, whereas 84 genes in the molecular functions were involved in binding (including ATP, DNA, RNA, protein, and iron) and catalytic activity. Compared to the other two categories, 68 identified genes participated in a wide variety of biological processes. The main categories represented were DNA repair; the modification, transcription, and developmental processes of RNA; the response to an external biotic stimulus; the oxidation–reduction process; signal transduction and so on.

Based on gene annotation, 15 genes were considered as *a priori* candidate genes relevant to seed vigor because they are involved in either the stability of the genetic materials or cell redox homeostasis (Table 4). The functions of these genes associated with DNA binding (*Glyma.06G027200*, *Glyma.06G143400*, *Glyma.06G147100*, *Glyma.06G217400*, and *Glyma.16G104400*), DNA repair (*Glyma.06G216300* and *Glyma.16G114600*), and the oxidation–reduction process (*Glyma.06G145300*, *Glyma.06G215100*, *Glyma.06G215200*, *Glyma.15G243200*, *Glyma.16G110700*, *Glyma.16G111300*, *Glyma.16G113000*, and *Glyma.16G113100*).

**Table 4.** Profiles of the most promising candidate genes for seed vigor in soybean.

| Candidate Genes | Description | GO Term [1] |
|---|---|---|
| *Glyma.06G027200* | SRF-type transcription factor (DNA binding and dimerisation domain) | GO-MF:0000977 RNA polymerase II regulatory region sequence-specific DNA binding |
| *Glyma.06G143400* | Mini-chromosome maintenance replisome factor | GO-MF:0003682 chromatin binding |
| *Glyma.06G147100* | WRKY DNA binding domain | GO-MF:0003700 sequence-specific DNA binding transcription factor activity |
| *Glyma.06G217400* | DNA-dependent RNA polymerase | GO-BP:0006351 transcription, DNA-templated |
| *Glyma.16G104400* | DNA replication licensing factor MCM3 | GO-MF:0003677 DNA binding |
| *Glyma.06G216300* *Glyma.16G114600* | DNA helicase PIF1/RRM3; PIF1-like helicase | GO-BP:0006281 DNA repair GO-BP:0006974 cellular response to DNA damage stimulus |
| *Glyma.06G145300* | Peroxidase | GO-BP:0006979 response to oxidative stress GO-BP:0098869 cellular oxidant detoxification GO-MF:0016491 oxidoreductase activity |
| *Glyma.06G215100* | Thioredoxin | GO-BP:0045454 cell redox homeostasis GO-MF:0003756 protein disulfide isomerase activity |
| *Glyma.06G215200* *Glyma.15G243200* *Glyma.16G110700* | Cytochrome P450 | GO-BP:0055114 oxidation–reduction process GO-MF:0016705 oxidoreductase activity, acting on paired donors, with incorporation or reduction of molecular oxygen |
| *Glyma.16G111300* | Iron/ascorbate family oxidoreductase | GO-BP:0055114 oxidation-reduction process GO-MF:0016706 oxidoreductase activity, acting on paired donors, with incorporation or reduction of molecular oxygen, 2-oxoglutarate as one donor, and incorporation of one atom each of oxygen into both donors |
| *Glyma.16G113000* | Hydroxymethylglutaryl-coenzyme A reductase | GO-BP:0055114 oxidation–reduction process GO-MF:0016616 oxidoreductase activity, acting on the CH-OH group of donors, NAD or NADP as acceptor |
| *Glyma.16G113100* | Thioredoxin reductase | GO-BP:0019430 removal of superoxide radicals GO-MF:0004791 thioredoxin-disulfide reductase activity |

[1] BP, Biological process; MF, Molecular function.

## 4. Discussion

Due to their high lipid content, soybean seeds generally have short lives compared to many other crops and the post-harvest deterioration is much more acute under tropical climatic regions [39]. Poor vigor irreversibly affects seed quality which further leads to the failure of field emergence and seedling establishment after germination. To date, studies regarding seed vigor are still relatively backward in soybean, though a number of candidate genes have been already published for many other species [40–50]. Genetic studies on seed vigor have led to useful analyses in relation to the acceleration of crop gene discovery; however, a key problem is that using conventional molecular markers limits the accuracy of QTL detection [51–54]. With decreasing costs and increasing capacity in relation to sequencing technology, RAD marker sequencing offers a viable and cost-effective option to identify promising candidate genes or sequence variants for seed vigor.

### 4.1. An Accurate Phenotyping of Seed Vigor Requires Multiple Indicators

As natural aging requires a long period of time, the seeds were generally treated with high temperatures and a high relative humidity to artificially accelerate deterioration. For *Glycine max*, the International Seed Testing Association (ISTA) [1] has defined very specific protocols using the AA test. Consistent with many seed lots, the germination proportion of soybean during storage exhibits a lag period which is essentially constant before a relatively rapid reduction [55]. When a seed lot is close to the tipping point, it may appear to have poor emergence in the field but high germination in the laboratory-based condition [11,21]. Besides, seed vigor is a complex quantitative trait that integrates both germination ability and seedling characteristics [23,56]. It should be noted that the prediction of seed vigor is elusive, as the deterioration of soybean may be underestimated by simply utilizing seed germination (radicle emerging from seed coat more than 5 mm) as the indicator of viability. The seedling root, shoot length, seedling fresh weight, and dry weight of seeds from less vigorous cultivars were substantially reduced with the prolonging of ambient storage. In contrast, the seedling performance of vigorous soybean cultivars exhibited minor influences from natural aging [57,58]. Thus, only using germination test results (normally germination percentage) prevents an evaluation of the post-harvest seed deterioration effectively; other traits regarding seedling performance (e.g., shoot length) should also be considered.

However, in most of the 'seed vigor' literature for soybean, total germination proportion is the only/major trait determined [16–19]. In addition, weak correlations between germination ability and seedling performance were often found in the investigation of seed vigor [59–61], though the germination rate appeared to be closely associated with seedling-related traits ($R^2 > 0.5$) in some research [13,18].

In this study, we carried out a range of assessments for seed germination performance and seedling growth characteristics, and the relevant indicators were calculated as well. There were no significant correlations between germination and seedling growth traits under both natural and accelerated aging treatments (Figure 2 and Table S2). Meanwhile, various QTLs were identified depending on the parameters used for the analysis (Table 2), which indicated that several genes were likely to be involved in the determination of seed vigor. It is therefore important for the accurate phenotyping of seed vigor that several different traits are measured.

### 4.2. Mechanisms Conferring the Loss of Seed Vigor in Soybean Might Be Different under −20 °C Storage and Accelerated Aging Conditions

Considerable variation in seed vigor is a genetic characteristic which is largely influenced by seed quality, seed moisture content, storage temperature, relative humidity, and biotic factors, and leads to differential speeds of seed deterioration among soybean varieties [2,4]. For practical reasons, breeders/scientists commonly rely on artificial aging and germination assays to predict seed vigor. According to the proteome analyses in *Arabidopsis*, Rajjou et al. [62] proposed that the CD protocol truly mimics the deterioration

process during natural aging due to the common features between the artificially and naturally aged seeds.

There is increasing evidence for several species, including soybean, that the accelerated aging test does not completely (or even fails to) simulate the effect of natural aging [13,63–70]. Studies with soybeans found varied responses in free radical levels [71,72] and a low correlation between the performance [13] under ambient and artificial aging conditions. Results of *Atriplex cordobensis* seeds showed that conductivity and malondialdehyde (MDA) assays were sensitive to accelerated deterioration but not natural aging [64]. In agreement with earlier points, Murthy et al. [69] showed that the viability loss of *Vigna radiata* seeds varied under different storage conditions as the result of differing contributions of primary biochemical reactions. Additionally, lettuce seed germination was poorly correlated between conventional storage and controlled deterioration conditions and the same QTLs were not identified as well [67]. Different patterns of active oxygen species levels and antioxidative enzyme activities in neem (*Azadirachta indica*) seeds exposed to natural aging and controlled deterioration revealed the possibility of multiple aging pathways [66].

In the present study, seed samples of soybean showed a more rapid loss of viability at a high temperature (41 °C) and a relatively high humidity (99%) even for a very short period (3 d) in comparison with storage at −20 °C for 5 years. A reduction in the germination rate from 88% to 55% for ZH24 and from 100% to 96% for HX3 was observed under AA conditions, whereas both soybean cultivars were found to remain stable (above 95%) under storage at −20 °C (Table 1, Figure 1E). On the contrary, seedling growth (SL and SFW) of the soybean cultivar (ZH24) with poor vigor was more obviously affected under cold temperature during the long storage time rather than at high temperature for a short period (Table 1, Figure 1A,C). Our results suggest that dissimilar events occurred during natural and accelerated aging, as higher temperatures and relative humidity produced more severe impairments in germination, whereas the aging-related impacts on seedling establishment were mainly attributed to the length of storage time. Additionally, the complete lack of correlations between all parameters under the −20 °C and AA storage conditions (Figure 2, Table S2), and the general failures to identify the overlapped QTL when the 2013 and 2018 RIL populations were compared (Table 2), indicate that the mechanisms underlying the loss of germinability, and the retardation of seedling growth vary during aging. The conclusion is that there is promising evidence for the genetic basis underlying the ability to germinate acting independently of seedling performance after germination.

Seed aging is affected distinctly by different moisture levels and temperatures. The findings of the present study underline the complexity and polygenic nature of seed vigor for soybean. We thereby speculate that different deterioration mechanisms may be involved in −20 °C conservation and artificial aging storage. The divergent deterioration of soybeans under natural and accelerated aging conditions might be due to the differences in physical [73], physiological (decreased respiratory activity and membrane permeability) [74], and biochemical alterations [75] that irreversibly accumulate during storage. Chemical changes probably happened in the lipid and protein molecules of soybeans stored at high relative humidity and temperature (here 41 °C, 99% RH). The natural aging conditions used soybean seeds conserved with moisture levels below 20% and a temperature of −20 °C, and these seeds might have remained in the metabolically inactive state. Studies have found that seeds undergoing some deterioration had a lower in vivo metabolic activity than seeds without deterioration [76]. In addition to storage conditions, the growing conditions of seed-producing plants clearly have a major effect on seed vigor as well [77–79]. Therefore, a better understanding of the physiological and molecular characteristics in seeds from the same growth conditions exposed to natural and accelerated aging conditions is still needed.

### 4.3. Fifteen Candidate Genes Associated with the Seed Vigor of Soybean

Genetic mapping is an effective approach to identify genomic regions and functional variations in genes controlling a given trait in crops [54,80]. So far, there have been 18 seed vigor-related QTLs detected in soybean, and no common QTL was found among the different research populations [17–19]. Due to technical limitations, information regarding candidate genes near/within the identified SSR markers/QTL intervals was missing in the earlier mapping studies. Only in 2019 did Zhang et al. [18] identify 19 candidate genes related to seed development, seed germination, seed dormancy, seed coat formation, fatty acid/lipid metabolic process, and seed vigor after storage in soybean by using high-density linkage maps of two RIL populations (LM6 and ZM6). In this study, a more comprehensive investigation (including nine parameters of germination and seedling characteristics) for seed vigor in soybean was conducted, and five and three QTL hotspots were identified under −20 °C conservation and accelerated aging storage, respectively. Among those 32,616 unique genomic variants between ZH24 and HX3, 220 significant variants probably form functional mutations in 117 novel genes. Finally, according to the annotation of the GO enrichment analysis, we obtained 15 highly promising candidate genes for the seed vigor of soybean.

During storage, seeds undergo irreversible "aging processes" and/or "deterioration events" which result in delayed seed germination and poor seedling establishment, often accompanied with high levels of reactive oxygen species and increasing damage of DNA, RNA, proteins, etc., in aged seeds [81–83]. Regarding soybean, previous research has reported several aspects including DNA and RNA stability, ROS scavenging, and lipid metabolism that are relevant to seed deterioration and vigor [55,84–88]. The ability of aged soybean seeds to germinate decreases markedly as the storage time increases, and the loss of germination potential was tracked by a decline in RNA integrity in soybean seeds after 17 years of storage [55]. Besides, an in-depth proteomic analysis of *Glycine max* seeds revealed that the decreased abundance of proteins associated with primary metabolism, ROS detoxification, translation elongation and initiation, protein folding, and proteolysis during the controlled deterioration treatment, and the accumulation of $H_2O_2$ and MDA were observed as well [86]. Free radical and lipid peroxidation were considered to be major contributors to seed deterioration in soybean [88]. Studies have demonstrated that phospholipase Dα (*PLDα*) affecting seed phospholipid and triacylglycerol profiles reduced the seed viability of soybean. The suppression of *PLDα* activity in soybean seed accompanied seed vigor enhancement during the natural aging process [85]. Similarly, *PLDα1*-knockdown soybean seeds displayed higher unsaturated glycerolipid contents and seed vigor when grown in high temperature and humidity environments [84]. The conversion of triacylglycerol to soluble sugars was proven to be important for the germination and seedling establishment of aged seeds of soybean [87]. In addition, a glutathione S-transferase-interacting annexin of soybean, *GmANN* (*Glyma.13G088700*), is involved in enhanced seed vigor under high temperature and humidity stress in *GmANN*-transgenic *Arabidopsis* seeds [49].

For this study, seven candidate genes encoded DNA-interacting proteins, while the other eight candidates coded for enzymes involved in the oxidation-reduction process (Table 4). Among the proteins that interact with DNA, SRF-type transcription factor (*Glyma.06G027200*), mini-chromosome maintenance replisome factor (*Glyma.06G143400*), WRKY DNA-binding domain (*Glyma.06G147100*), and DNA replication licensing factor MCM3 (*Glyma.16G104400*) were annotated to DNA binding. DNA-dependent RNA polymerase encoded by *Glyma.06G217400* uses DNA templates for RNA synthesis during transcription. DNA helicase PIF1/RRM3 playing an essential role in DNA replication, recombination, and repair is encoded by two genes *Glyma.06G216300* and *Glyma.16G114600*. Furthermore, peroxidase (*Glyma.06G145300*) contributes to cellular oxidant detoxification by catalyzing the compound oxidation. Thioredoxin (*Glyma.06G215100*), cytochrome P450 (a monooxygenase, *Glyma.06G215200*, *Glyma.15G243200*, and *Glyma.16G110700*), iron/ascorbate family oxidoreductase (*Glyma.16G111300*), hydroxymethylglutaryl-coenzyme A reductase

(*Glyma.16G113000*), and thioredoxin reductase (*Glyma.16G113100*) are associated with oxidoreductase activity against various oxidative stresses.

## 5. Conclusions

Collectively, characterizing the seed vigor-related genes will provide a better understanding of the genetic and regulatory mechanisms of seed vigor in soybean, which may enhance the seed vigor by breeding. A set of 165 recombinant inbred lines (RILs) derived from a cross between ZH24 and HX3 were used to detect QTLs related to seed vigor after post-harvest storage by constructing a high-density linkage map. In total, forty-eight QTLs for nine seed vigor-related parameters under NA and AA conditions were mapped on chromosomes 1, 3, 4, 5, 6, 7, 8, 9, 11, 12, 13, 15, 16, 17, 19, and 20. However, no loci detected under −20 °C storage overlayed with loci from accelerated aging treatment, suggesting that different mechanisms might be involved in seed deterioration under two storage conditions. Besides, the germination-related traits were significantly correlated with each other under the same conditions, similar to seedling establishment parameters SFW, SVI, and VI. It was not surprising that the QTL hotspots for germination ability (GP, GI; GR, NSR, and GI) and seedling performance (SFW, SVI, and VI) were observed, respectively. Based on the GO annotation in major QTLs and QTL hotspots, we selected 15 candidate genes regarding DNA binding, DNA repair, and the oxidation–reduction process.

Seed vigor is a genetic trait with significant diversity among soybean germplasms. Systematic tests on different populations, or a genome-wide association study on a larger natural population would enable the discovery and characterization of new genetic variants associated with seed vigor in *Glycine max*. Besides, seed vigor has complex genetic mechanisms during the prolonged storage period. Therefore, further functional analysis is still needed to validate putative candidate genes. Meanwhile, the integration of multiple '-omics' approaches revealing molecular mechanisms of seed vigor, will help to make a significant breakthrough in the breeding of soybean varieties with superior germination and seedling growth.

**Supplementary Materials:** The following are available online at https://www.mdpi.com/article/10.3390/cimb43030136/s1.

**Author Contributions:** R.W. and C.Y. conceived and designed the research. R.W. analyzed data, performed most of the bioinformatic analysis, and wrote the manuscript. F.W. conducted most of the lab experiments. X.X. contributed with genetic comparison of parental cultivars. C.Y. incorporated necessary corrections or modifications within the manuscript. All authors have read and agreed to the published version of the manuscript.

**Funding:** This work was supported by the Major Program of Guangdong Basic and Applied Research (2019B030302006), the National Key Research and Development Program of China (2018YFD0100901), and the National Key Project for Research of Seven Major Crop Breeding (2016YFD0101901), and the Guangdong Science and Technology Program Project (2016A030303051).

**Informed Consent Statement:** Not applicable.

**Data Availability Statement:** The datasets of RAD-seq for this study can be found in NCBI 181(SRP065356).

**Acknowledgments:** We thank all group members for the aid of plant materials preparation, and special appreciation to Wenting Li for her great help regarding bioinformatic analysis.

**Conflicts of Interest:** The authors declare no conflict of interest.

**Abbreviations**

| | |
|---|---|
| AA | Accelerated aging |
| ADD | Additive effect |
| Chr. | Chromosome |
| CIM | Composite interval mapping |
| cv. | Cultivar |
| HX3 | Huaxia3hao |
| GI | Germination index |
| GO | Gene ontology |
| GP | Germination potential |
| GR | Germination rate |
| InDel | Insertion/Deletion |
| LOD | Logarithm of odds |
| NA | Natural aging |
| NSR | Normal seeding rate |
| PVE | Percentage of phenotypic variation explained |
| QTL | Quantitative trait locus |
| RAD-seq | Restriction site-associated DNA sequencing |
| RH | Relative humidity |
| RIL | Recombinant inbred line |
| RL | Root length |
| SFW | Seedling fresh weight |
| SL | Seedling length |
| SNP | Single nucleotide polymorphism |
| SSD | Singe seed decent |
| SVI | Simple vigor index |
| VI | Vigor index |
| ZH24 | Zhonghuang24 |

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
