# Peer review of "Quantitative Trait Locus Mapping of Seed Vigor in Soybean under −20 °C Storage and Accelerated Aging Conditions via RAD Sequencing"

_cimb, doi:10.3390/cimb43030136_

Round 1

Reviewer 1 Report

This is an interersting presentation on an important topic. It requires revision for grammar, and to some extent, organization. The authors shouold make it clear in ythe early p-art of the Introduction exactly how they are defining 'seed vigor'.  With appropriate revision it represents a significant contribution.   

Author Response

Response to Reviewer 1:

Q1: This is an interesting presentation on an important topic. It requires revision for grammar, and to some extent, organization. The authors should make it clear in the early part of the Introduction exactly how they are defining 'seed vigor'.  With appropriate revision it represents a significant contribution.

A1: Seed vigor is a complex physiological trait, reflecting comprehensive potentials of seed germination and normal seedling establishment under a wide range of adverse and stressful conditions, such as high temperature and moisture.

Reviewer 2 Report

Seed vigor is regulated by a very complicated network of signaling and gene expression in soybean, and its molecular mechanisms are still largely unknown. The present study utilized soybean seeds of a recombinant inbred line (RIL) population derived from the cross between a low vigor cultivar (ZH24) and a vigorous cultivar (HX3) to identify quantitative trait loci (QTLs) underlying seed vigor under −20°C and accelerated aging condition. The authors identified seed vigor-related QTLs under both −20°C and accelerated aging storages. Their QTL analysis indicated that deterioration under accelerated aging conditions is not correlated to long storing at −20°C. Further, the study proposed 15 candidate genes for possible roles in seed vigor of soybean. The subject matter falls within the scope of journal. The paper shows an original contribution in the area of soybean genetics and breeding. The QTL methodology seems appropriate and the text flow is well organized.

Hereafter, some specific comments for the authors to consider.

  1. There were quite a number of markers which could not be linked to any other marker. If I am correct, a total of 52306 SNP markers were generated but only 5425 were mapped (9.6%). This is unexpected as the number of markers and progenies used are enough to get a denser map. This may point to large number of missing values (technical problems encountered during analysis) or considerable genotypic errors. No explanation is provided.

In addition, the average distance between adjacent markers was not so low. Current available maps in soybean are far below the 1.46 cM average density. The total number of markers mapped, unmapped (unlinked) and the average marker-interval range should be mentioned in the main text.

  1. Seed vigor is considered to be a genetic trait with significant diversity among soybean germplasms. This study is based on the germplasm of two cultivars. So, the identified QTL should be considered as preliminary for breeding seed vigor in soybean. Further systematic tests on other populations are necessary.

  1. In my opinion the most interesting part of the work is the assessment of QTLs under two conditions. This provides the possibility to the authors to make valuable statement about QTL stability over AA or NA storage. The results suggest that the genetic mechanism of seed aging is different under divergent deterioration treatments. Little is discussed about this.
  2. The authors stated that: "based on gene annotation, 15 genes were considered as a priori candidate genes relevant to seed vigor". This is unclear. The authors need to provide more details on how the 15 genes were selected.
  3. Authors stated in conclusion: "Collectively, this work contributes to a better insight into genomic mechanisms controlling seed vigor in Glycine max, and identified several candidate genes ...". In my view, no any new insight (or very little) into genetic mechanism is shown and the proposed candidate genes are just a hypothesis. The authors need further functional analysis to validate the candidate genes. Characterizing the seed vigor-related genes will provide a better understanding of the genetic and regulatory mechanisms of seed vigor in soybean, which may allow to enhance seed vigor by breeding. This has to be clarified in the conclusions.
  4. The authors indicate statistical differences in Table 1 and S2. However, they do not explain how they obtained the p-values. n values (RIL, parents) are not provided in Figure 1 & Table 1. No statistical significance analysis provided for Figure 2.
  5. The authors should provide in the very beginning the ploidy of the soybean cultivars crossed. This may be helpful for readers outside the research field of soybean.
  6. I would prefer the "conclusion part" before the M & Methods.

Author Response

Response to Reviewer 2:

Seed vigor is regulated by a very complicated network of signaling and gene expression in soybean, and its molecular mechanisms are still largely unknown. The present study utilized soybean seeds of a recombinant inbred line (RIL) population derived from the cross between a low vigor cultivar (ZH24) and a vigorous cultivar (HX3) to identify quantitative trait loci (QTLs) underlying seed vigor under −20°C and accelerated aging condition. The authors identified seed vigor-related QTLs under both −20°C and accelerated aging storages. Their QTL analysis indicated that deterioration under accelerated aging conditions is not correlated to long storing at −20°C. Further, the study proposed 15 candidate genes for possible roles in seed vigor of soybean. The subject matter falls within the scope of journal. The paper shows an original contribution in the area of soybean genetics and breeding. The QTL methodology seems appropriate and the text flow is well organized.

Hereafter, some specific comments for the authors to consider.

Q1: There were quite a number of markers which could not be linked to any other marker. If I am correct, a total of 52306 SNP markers were generated but only 5425 were mapped (9.6%). This is unexpected as the number of markers and progenies used are enough to get a denser map. This may point to large number of missing values (technical problems encountered during analysis) or considerable genotypic errors. No explanation is provided.

In addition, the average distance between adjacent markers was not so low. Current available maps in soybean are far below the 1.46 cM average density. The total number of markers mapped, unmapped (unlinked) and the average marker-interval range should be mentioned in the main text.

A1: Sorry for the confusion. Based on 0.2 × RAD-seq of the RIL population, a total of 52306 SNP sites/markers were detected. But some markers are redundant, meaning that they are completely correlated or identical in the population. So, redundant markers were integrated into a recombination bin unit, and 5425 recombinant bins were obtained from 52306 SNP sites/markers in our study. Based on the genotypes of 5425 bins, a high-density bin linkage map was constructed covering 7932.8 cM, with an average distance of 1.46 cM between adjacent bin markers.

Thanks for asking, we made this point clearer in section 2.2 Identification of QTL Determining Seed Vigor of Soybean.

Q2: Seed vigor is considered to be a genetic trait with significant diversity among soybean germplasms. This study is based on the germplasm of two cultivars. So, the identified QTL should be considered as preliminary for breeding seed vigor in soybean. Further systematic tests on other populations are necessary.

A2: Yes, for sure, two cultivars are far from enough to make a full picture of genetic mechanisms for seed vigor. Therefore, we have applied GWAS (Genome-Wide Association Study) analysis on a germplasm collection, which consisted of more than 300 soybean cultivars. The work is still ongoing.

Q3: In my opinion the most interesting part of the work is the assessment of QTLs under two conditions. This provides the possibility to the authors to make valuable statement about QTL stability over AA or NA storage. The results suggest that the genetic mechanism of seed aging is different under divergent deterioration treatments. Little is discussed about this.

A3: Thanks for this nice recommendation, we discussed this in section 3.2. Mechanisms Conferring the Loss of Seed Vigor in Soybean Might be Different under 20°C Storage and Accelerated Aging Condition as follows.

Seed aging is affected distinctly by different moisture levels and temperatures. The findings of the present study underline the complexity and polygenic nature of seed vigor for soybean. We thereby speculate that different deterioration mechanisms may be involved in −20°C conservation and artificial aging storage. The divergent deterioration of soybeans under natural and accelerated aging conditions might be due to the differences in physical [66], physiological (decreased respiratory activity and membrane permeability) [67], biochemical alterations [68] that irreversibly accumulate during storage. Chemical changes probably happened in the lipid and protein molecules of soybeans stored at high relative humidity and temperature (here 41°C, 99% RH). The natural aging condition used soybean seeds conserved with moisture below 20% and temperature at −20°C, and these seeds might remain in the metabolically inactive state. Studies found that seeds undergone some deterioration had a lower in vivo metabolic activity than seeds without deterioration [69]. In addition to storage condition, the growing conditions of seed-producing plants clearly have a major effect on seed vigor as well [70-72]. Therefore, a better understanding of the physiological and molecular characteristics in seeds from the same growth conditions exposed to natural and accelerated aging conditions is still needed.

Q4: The authors stated that: "based on gene annotation, 15 genes were considered as a priori candidate genes relevant to seed vigor". This is unclear. The authors need to provide more details on how the 15 genes were selected.

A4: The selection details are as follows “Based on gene annotation, 15 genes were considered as a priori candidate genes relevant to seed vigor, because they are involved in either stability of genetic materials or cell redox homeostasis (Table 4).”

Additionally, “During deterioration, a range of irreversible metabolic and cellular alterations including oxidation of lipid, protein and nucleic acids, enzyme inactivation, membrane perturbations, and impairment of DNA, RNA and protein biosynthesis generally occur in aged seeds [2, 4, 23]. In this regard, the levels of seed vigor are considered to be associated with the balance between oxidative damages and self-protective as well as repair mechanisms such as antioxidant systems [24].” mentioned in the introduction. Therefore, these 15 candidate genes were selected based on gene functions related to seed vigor.

Q5: Authors stated in conclusion: "Collectively, this work contributes to a better insight into genomic mechanisms controlling seed vigor in Glycine max, and identified several candidate genes ...". In my view, no any new insight (or very little) into genetic mechanism is shown and the proposed candidate genes are just a hypothesis. The authors need further functional analysis to validate the candidate genes. Characterizing the seed vigor-related genes will provide a better understanding of the genetic and regulatory mechanisms of seed vigor in soybean, which may allow to enhance seed vigor by breeding. This has to be clarified in the conclusions.

A5: Thanks a lot for this comment. We clarified this point as below which is marked in yellow. And the suggestion from the second comment was also included at the end of the conclusions.

Conclusions

Collectively, characterizing the seed vigor-related genes will provide a better understanding of the genetic and regulatory mechanisms of seed vigor in soybean, which may allow enhancing seed vigor by breeding. A set of 165 recombinant inbred lines (RILs) derived from a cross between ZH24 and HX3 were used to detect QTLs related to seed vigor after post-harvest storage by constructing a high-density linkage map. In total, forty-eight QTLs for nine seed vigor-related parameters under NA and AA conditions were mapped on chromosomes 1, 3, 4, 5, 6, 7, 8, 9, 11, 12, 13, 15, 16, 17, 19, and 20. However, no loci detected under −20°C storage overlayed with loci from accelerated aging treatment, suggesting different mechanisms might be involved in seed deterioration under two storage conditions. Besides, germination-related traits were significantly correlated with each other under the same condition, so as seedling establishment parameters SFW, SVI and VI. Not surprising that QTL hotspots for germination ability (GP, GI; GR, NSR, and GI) and seedling performance (SFW, SVI, and VI) were observed respectively. Based on GO annotation in major QTLs and QTL hotspots, we selected 15 candidate genes regarding DNA binding, DNA repair, and the oxidation-reduction process.

Seed vigor is a genetic trait with significant diversity among soybean germplasms. Systematic tests on different populations, or genome-wide association study on a larger natural population would enable the discovery and characterization of new genetic variants associated with seed vigor in Glycine max. Besides, seed vigor has complex genetic mechanisms during the prolonged storage period. Therefore, further functional analysis is still needed to validate putative candidate genes. Meanwhile, the integration of multiple “-omics” approaches revealing molecular mechanisms of seed vigor, will help to make a significant breakthrough in the breeding of soybean varieties with superior germination and seedling growth.

Q6: The authors indicate statistical differences in Table 1 and S2. However, they do not explain how they obtained the p-values. n values (RIL, parents) are not provided in Figure 1 & Table 1. No statistical significance analysis provided for Figure 2.

A6:

  • Table 1 Student's t-test was used to test the significant differences between two parental cultivars (“ZH24” and ”HX3”).
  • Table S2 Spearman’s rank correlation was used to test the association between natural aging and artificial aging conditions. The correlation coefficient was calculated based on the phenotypic values of all the RILs.

The methods of calculation and relevant software for Table 1, Table S2, Figure 2 were described in more detail in section 4.6 Statistical analysis.

  • Figure 1 & Table 1: the number of RILs is 168, this information is added in both titles.
  • Figure 2: the results of statistical significance analysis were presented in Table S2.

Q7: The authors should provide in the very beginning the ploidy of the soybean cultivars crossed. This may be helpful for readers outside the research field of soybean.

A7: We added the ploidy of the soybean cultivars in the Introduction as follows.

“In this study, we used a RILs population developed from a cross between two diploid cultivars (cv.) ‘Zhonghuang24’ and ‘Huaxia3hao’.”

Q8: I would prefer the "conclusion part" before the M & Methods.

A8: Of course, presenting the conclusion part before the Materials & Methods seems more logical, but it is a pity that the Journal template puts Materials & Methods in front. We double-checked the newly published papers in this journal, it is also the same order.

Reviewer 3 Report

In this manuscript, author did the quantitative trait locus mapping of seed vigor in soybean under −20℃ storage and accelerated aging condition via RAD sequencing. In this study to analyze the genetic background of seed vigor, soybean seeds of a recombinant inbred line (RIL) population derived from the cross between Zhonghuang24 (ZH24, low vigor cultivar) and Huaxia3hao (HX3, vigorous cultivar) were utilized to identify quantitative trait loci (QTLs) underlying seed vigor under −20°C conservation and accelerated aging condition. The linkage analysis identified multiple seed vigor-related QTLs under both −20℃ and accelerated aging storages. Two major QTLs and eight QTL hotspots localizing on chromosomes 3, 6, 9, 11, 15, 16, 17, and 19 were detected associated with seed vigor across two storage conditions. However, the indicators of seed vigor did not correlate well between two aging treatments, and no common QTLs were detected in RIL populations stored in two conditions. These results indicate that deterioration under accelerated aging conditions is not reflective of natural aging at −20°C. Additionally, we suggest 15 promising candidate genes possibly determine the seed vigor in soybean, which would help explore mechanisms responsible for maintaining high seed vigor.

The manuscript is very well written and cover all the scientific aspect required to be published in this journal. I found no major fault in this manuscript. Still, for the betterment of this manuscript, I have 2 suggestion for the authors:

In Introduction change

  1. vigor seeds which are tolerant to vigor seeds that are tolerant.
  2. markers (Satt371, Satt453, and Satt618) identified for linkage to markers (Satt371, Satt453, and Satt618) were identified for linkage.
  3. primary process initiates seed aging to primary process that initiates seed aging.

Author Response

Response to Reviewer 3:

In this manuscript, author did the quantitative trait locus mapping of seed vigor in soybean under −20℃ storage and accelerated aging condition via RAD sequencing. In this study to analyze the genetic background of seed vigor, soybean seeds of a recombinant inbred line (RIL) population derived from the cross between Zhonghuang24 (ZH24, low vigor cultivar) and Huaxia3hao (HX3, vigorous cultivar) were utilized to identify quantitative trait loci (QTLs) underlying seed vigor under −20°C conservation and accelerated aging condition. The linkage analysis identified multiple seed vigor-related QTLs under both −20℃ and accelerated aging storages. Two major QTLs and eight QTL hotspots localizing on chromosomes 3, 6, 9, 11, 15, 16, 17, and 19 were detected associated with seed vigor across two storage conditions. However, the indicators of seed vigor did not correlate well between two aging treatments, and no common QTLs were detected in RIL populations stored in two conditions. These results indicate that deterioration under accelerated aging conditions is not reflective of natural aging at −20°C. Additionally, we suggest 15 promising candidate genes possibly determine the seed vigor in soybean, which would help explore mechanisms responsible for maintaining high seed vigor.

The manuscript is very well written and cover all the scientific aspect required to be published in this journal. I found no major fault in this manuscript. Still, for the betterment of this manuscript, I have 2 suggestions for the authors:

In Introduction change

Q1: vigor seeds which are tolerant to vigor seeds that are tolerant.

A1: We corrected the sentences as follows.

Those cultivars with better vigor seeds that are tolerant to adverse conditions during storage would be important germplasm resources for higher soybean yields.

Q2: markers (Satt371, Satt453, and Satt618) identified for linkage to markers (Satt371, Satt453, and Satt618) were identified for linkage.

A2: In addition, based on the seed germination performance of 33 soybean genotypes under ambient and AA storages, three SSR markers (Satt371, Satt453, and Satt618) were identified for linkage with both seed vigor and seed coat color.

Q3: primary process initiates seed aging to primary process that initiates seed aging.

A3: Hence, some researchers assume that the major primary process that initiates seed aging could be different under AA and NA conditions, and the controversy on whether mechanisms of seed aging are similar under different deterioration treatments continues.

Reviewer 4 Report

Rongfan Wang et al. present an interesting study devoted to the identification of QTL responsible for seed vigor in important agricultural plants, while they compared natural senescence and artificial senescence, induced by seed exposure to the conditions of increased temperature and humidity. Multiple parameters were used to evaluate seed vigor, and accurate phenotyping of seeds is an indisputable advantage of this work. Several QTLs were detected, and candidate genes for the seed vigor of soybean were selected. 16 of 465 identified variant genes have a high potential to produce a phenotypic difference. A very important result of the study is the confirmation of little correlation between naturally aged and artificially aged seeds. While naturally aged seeds of ZH24 variety displayed a severe decrease in seedling establishment parameters, the germination capacity was significantly reduced during artificial aging.

No serious shortcomings preventing the publication of the submitted material have been identified.

Soybean seeds are short-lived as compared to other crops, probably due to the fact that these seeds are very rich in lipids. Unfortunately, there is little discussion in the work of the processes associated with lipid modification during seed senescence, and such genes are not discussed among the genes potentially involved in the regulation of seed senescence, although according to the supplementary data, such genes were present in identified QTL. It would be desirable to add such information to the text of the manuscript. The same goes for the identified genes associated with plant hormones.

Minor comments:

In Introduction – Please, clarify what does precisely “seed quality” means?

It looks like the sentence “Therefore, maintaining high vigor of seed during post-harvest storage provides an essential ingredient for improving crop production, which is of both economic importance” is not complete.

Author Response

Response to Reviewer 4:

Rongfan Wang et al. present an interesting study devoted to the identification of QTL responsible for seed vigor in important agricultural plants, while they compared natural senescence and artificial senescence, induced by seed exposure to the conditions of increased temperature and humidity. Multiple parameters were used to evaluate seed vigor, and accurate phenotyping of seeds is an indisputable advantage of this work. Several QTLs were detected, and candidate genes for the seed vigor of soybean were selected. 16 of 465 identified variant genes have a high potential to produce a phenotypic difference. A very important result of the study is the confirmation of little correlation between naturally aged and artificially aged seeds. While naturally aged seeds of ZH24 variety displayed a severe decrease in seedling establishment parameters, the germination capacity was significantly reduced during artificial aging.

No serious shortcomings preventing the publication of the submitted material have been identified.

Soybean seeds are short-lived as compared to other crops, probably due to the fact that these seeds are very rich in lipids. Unfortunately, there is little discussion in the work of the processes associated with lipid modification during seed senescence, and such genes are not discussed among the genes potentially involved in the regulation of seed senescence, although according to the supplementary data, such genes were present in identified QTL. It would be desirable to add such information to the text of the manuscript. The same goes for the identified genes associated with plant hormones.

Thanks a lot for these constructive recommendations. I hope the genes you mentioned ‘present in identified QTL’ are not the genes shown in Table S4 and S5, because most of the variants are predicted not significant.

  1. Lipid modification during seed senescence. Yes, you are right. Due to high lipid contents, lipid modification/oxidation during seed senescence negatively impacts the seed vigor of soybean. This point is considered during candidate gene selection. But as shown in Table S6 and S7 “The Gene Ontology (GO) enrichment analysis of significant variants”. Genes that relate to lipids function in the lipid biosynthetic process (Glyma.06G215400,Glyma.06G214800); phospholipid binding (Glyma.06G223100); protein lipidation (Glyma.16G109400). It seems that they are not directly involved in lipid metabolism, that's why we did not consider them as candidate genes. Anyway, thanks to all the same. We will consider your nice comments in another study since there is good evidence that poor seed vigor of soybean is closely correlated with lipid modification/oxidation.
  2. Genes associated with plant hormones. Regarding plant hormones, gibberellin plays a role in seed germination. However, the gene detected in our study is Glyma.16G110600 coding for the gibberellin-regulated protein. It is not directly related to seed vigor.

Minor comments:

Q1: In Introduction – Please, clarify what does precisely “seed quality” means?

A1: The seed quality indicates initial condition of the seed. ‘It is controlled by quantitative genetic factors, the initial condition of the seed, biotic and abiotic factors during storage.’

Q2: It looks like the sentence “Therefore, maintaining high vigor of seed during post-harvest storage provides an essential ingredient for improving crop production, which is of both economic importance” is not complete.

A2: ‘Therefore, maintaining high vigor of seed during post-harvest storage provides an essential ingredient for improving crop production, which is of both economic and ecologic importance.’

Round 2

Reviewer 2 Report

The authors have addressed the concerns raised in the initial view and the present version I think is now ready for publication.

I wanted to mentioned that the authors mark the text additions (or revisions) as "deleted text".  I assume that their intention is to label and rather to delete the corresponding changes at the end. It was a bit confusing. The authors should clarify this.

Author Response

Sorry for the confusion, there is probably a software problem with these "labeled/marked corrections" since they are invisible in my version. So we include a PDF version for the 2nd revision to avoid this problem.

If a word version is required, please let us know. 
